**Data Availability Statement:** Data cannot be shared publicly because this is a multi-center

# Comparing psychological versus pharmacological treatment in emotional disorders: A network analysis

**Francisco Jurado-González**[1]*, **Francisco García-Torres**[1], **Alba Contreras**[2], **Roger Muñoz-Navarro**[3], **César González-Blanch**[4], **Leonardo Adrián Medrano**[5], **Paloma Ruiz-Rodríguez**[6], **Eliana M. Moreno**[1], **Carolina Pérez-Dueñas**[1], **Antonio Cano-Vindel**[7], **Juan A. Moriana**[1]

1 Department of Psychology, University of Cordoba/Maimonides Institute for Biomedical Research of Cordoba (IMIBIC)/Reina Sofia University Hospital, Cordoba, Spain, 2 University Catholique San Antonio of Murcia, Murcia, Spain, 3 Department of Psychology and Sociology, Faculty of Social and Human Sciences, University of Zaragoza, Teruel, Spain, 4 Mental Health Centre, Marqués de Valdecilla University Hospital—IDIVAL, Santander, Cantabria, Spain, 5 Faculty of Psychology, Siglo 21 University, De los Latinos, Cordoba, Argentina, 6 Castilla La Nueva Primary Care Centre, Health Service of Madrid, Fuenlabrada, Madrid, Spain, 7 Faculty of Psychology, Complutense University of Madrid, Madrid, Spain

* fran.jurado@uco.es

## Abstract

Transdiagnostic group cognitive behavioural therapy (TD-GCBT) is more effective in improving symptoms and severity of emotional disorders (EDs) than treatment as usual (TAU; usually pharmacological treatment). However, there is little research that has examined the effects of these treatments on specific symptoms. This study used Network Intervention Analysis (NIA) to investigate the direct and differential effects of TD-GCBT + TAU and TAU on specific symptoms of anxiety and depression. Data are from a multicentre randomised clinical trial ($N$ = 1061) comparing TD-GCBT + TAU versus TAU alone for EDs. The networks included items from the PHQ-9 (depression) and GAD-7 (anxiety) questionnaire and mixed graphical models were estimated at pre-treatment, post-treatment and 3-, 6- and 12-month follow-up. Results revealed that TD-GCBT + TAU was associated with direct effects, mainly on several anxiety symptoms and depressed mood after treatment. New direct effects on other depressive symptoms emerged during the follow-up period promoted by TD-GCBT compared to TAU. Our results suggest that the improvement of anxiety symptoms after treatment might precipitate a wave of changes that favour a decrease in depressive symptomatology. NIA is a methodology that can provide fine-grained insight into the likely pathways through which treatments exert their effects.

## Introduction

Emotional disorders (EDs; mainly, anxiety and depressive disorders) are highly prevalent, with approximately 4.4% of the world's population suffering from a depressive disorder and 3.6% from an anxiety disorder [1]. The COVID-19 pandemic increased the prevalence of anxiety

Randomized Clinical Trial with medication (No. EUDRACT: 2013-001955-11 and Protocol Code: ISRCTN58437086) promoted by the Psicofundación and approved by the Corporate Clinical Research Ethics Committee of Primary Care of Valencia (CEIC- APCV) (as the national research ethics committee coordinator) and the Spanish Medicines and Health Products Agency. The CEICAPCV have some availability restrictions, as a part of the legal and ethical control data of a Randomized Clinical Trial with medication. Data are available from the promoter (Spain) for researchers who meet the criteria for access to confidential data. Contact via Psicofundación (Spanish Foundation for the Promotion, Scientific and Professional Development of Psychology). Address: Calle Conde de Peñalver, 45, 5o izquierda, 28006 Madrid, Spain.) for researchers who meet the criteria for access to confidential data.

**Funding:** This research was financially supported by Spanish State Research Agency. Ref. RTI2018-099201-B-C21, awarded to Antonio Cano-Vindel and ref. PID2019-107243RB-C22, awarded to Juan Antonio Moriana. The funders had no role in study design, data collection and analysis, decision to publish, or preparation of the manuscript.

**Competing interests:** The authors have declared that no competing interests exist.

and depression worldwide [2, 3]. Specifically, depressive disorders increased by 27.6% and anxiety disorders by 25.6% [3]. On the other hand, it is estimated that one third of primary care (PC) consultations could be related to the presence of EDs [4]. This implies significant economic costs for public health systems, as well as an increase in the negative effects associated with these disorders (reduced quality of life, disability, increased comorbidity, etc.) for the individual; effects aggravated by the pandemic caused by SARS-CoV-2 [5–7]. Despite this serious public health problem, EDs are often incorrectly identified and treated in PC. Because most people experience symptoms of depression at some time in their lives that might be part of normal experiences, they may receive inappropriate diagnoses and treatment [8, 9].

Cognitive behavioural therapy (CBT) is the treatment of choice to address EDs [10, 11]. Previous studies have demonstrated the equality or superiority of CBT over pharmacological interventions for the treatment of EDs as they show better long-term outcomes and lower risk of relapse [12–16]. Despite the importance of applying evidence-based interventions [17–19] and patients' preference for psychological intervention [20], pharmacological prescription is the usual treatment (TAU) in the PC setting [21]. This could be explained by the short duration of consultations, the poor training of general practitioners in psychological assessment and treatment, the high rates of comorbidity and the variety of specific protocols that prevent a comprehensive response to comorbid problems, among other reasons [22–24].

These difficulties have led to the development of alternative approaches such as transdiagnostic therapies, which are useful for simultaneously treating several EDs through integrated protocols [25]. Specifically, several studies and meta-analyses have shown that transdiagnostic CBT (TD-CBT) compared to no treatment or TAU is more effective in reducing clinical symptomatology, treating comorbidity and decreasing relapses, and also produces fewer side effects and reduces economic costs [26–29]. Furthermore, TD-CBT is effective in PC settings in both individual and group formats [27, 30], as well as in an abbreviated format [31]. However, in addition to taking into consideration the efficacy of a treatment, it is important to know the pathways of action through which therapeutic change occurs, as the identification of these pathways could facilitate the development of specific and more tailored treatments [32]. In this regard, the network analysis methodology has shown to be a useful technique to examine the differential effects of treatments on specific symptoms [33]. Network methodology is a set of analytical techniques derived from network theory where mental problems are conceptualised as a complex system of interactions between symptoms that influence each other [34]. It is important to note that the best interventions do not work for all ED patient profiles, so the use of new research techniques such as network analysis based on network methodology could guide us towards the development of more personalised treatment strategies [35]. However, most previous studies have compared the efficacy of treatments by noting whether there are differences in global scores, without providing information on the direct and differential effects of treatments on specific symptoms [26, 27, 36].

These relationships can be visualised in a network structure, where the nodes (i.e., circles) represent the observable variables (elements such as symptoms and/or other clinically relevant variables) and the lines linking them (i.e., edges) represent the association between the variables [37]. Network Intervention Analysis (NIA) is a recent tool that allows estimating a network of symptoms and observing which of them are directly affected by treatment over time, as well as analysing changes in the way symptoms combine [33]. NIA offers some advantages over traditional methods. For example, traditional research uses the sum of scores as an index of severity and provides results in terms of response vs. non-response [38]. In contrast, NIA allows using data from randomised controlled trials and including treatment assignment (e.g., experimental vs. control) as a node in the network and identifying differential effects of treatments on symptoms over time [39]. Moreover, as Mullarkey pointed out [40], NIA also allows

looking at indirect treatment effects (i.e., whether the change in one symptom is mediated by the modification of another). A previous study has used this methodology to compare the differential effects of combination therapy versus psychotherapy for the treatment of depressive symptoms, and showed the superiority of combination therapy in the treatment of some symptoms such as feeling entrapped and emotional lability [41]. Another study found that insomnia-oriented CBT is useful for indirectly improving depressive symptoms through direct effects on two main symptoms of insomnia: *early morning awakening* and d*ifficulty maintaining sleep* [33].

Although significant efforts have been made to compare efficacy between different interventions, it is also important to provide additional findings that might help to understand the pathways of action of treatment. In our study, we used data from a randomised clinical trial comparing TAU (pharmacological treatment) with TD-GCBT plus TAU [42]. Briefly, the results of this clinical trial revealed that TD-GCBT plus TAU was highly effective in reducing symptoms of EDs after treatment and at 3-, 6- and 12-month follow-up [36]. Although uncertainties remain regarding the specific impact of these treatments at symptom level and how each treatment may directly impact specific symptoms, the results obtained using NIA could provide complementary insight as it allows modifying the relationships between symptoms in the network in different ways. Therefore, this paper hypothesises that each treatment will have specific pathways of action. To investigate these differential effects of treatments, we set the following objectives: (a) to explore whether each treatment (e.g. TD-GCBT vs. pharmacological) has a different impact on symptom-symptom interaction; (b) to examine the effects of both interventions on the association between symptoms over time; and (c) to observe whether there are changes in the association between symptoms at different time points, since the direct effect on one symptom may trigger changes in others indirectly.

## Methods

### Study design and procedure

A secondary analysis was performed using data from the PsicAP clinical randomised controlled trial (see registration in http://www.isrctn.com/ISRCTN58437086; main outcomes in the following study [36]).

Participants were recruited from 22 primary care centres in eight regions of Spain by their general practitioners during a routine clinical visit, between the months of January and July 2016. The patients presented signs or mild-moderate clinical symptoms of depression, anxiety and/or somatoform disorders. Individuals receiving treatment with antidepressants, anxiolytics and/or hypnotics were also invited to participate by their general practitioners. After receiving information about the study, those willing to participate signed a written informed consent and responded to a set of questionnaires designed to determine if they were eligible to participate in the study. The self-administered questionnaires included measures of clinical symptoms that were evaluated using the Patient Health Questionnaire (PHQ) [43]. Specifically, the Patient Health Questionnaire-9 (PHQ-9) [44] and the Generalized Anxiety Disorder-7 (GAD-7) were used to assess symptoms of depression and anxiety, respectively [45]. Validated cut-off criteria ($\geq$10) were used to confirm the presence of symptoms suggestive of an ED. Patients with PHQ scores indicative of severe depression (PHQ-9 $\geq$ 20) and/or severe anxiety (GAD-7 $\geq$ 15) were re-evaluated by a clinical psychologist using a Structured Clinical Interview for DSM Axis-I Disorders (SCID-I), and were referred back to their primary care physician for referral to specialty care services. The inclusion criteria were: (a) age 18–65 years; (b) patients with mild to moderate EDs (anxiety, depression and/or somatisation); and (c) willingness to voluntarily participate in the study. The exclusion criteria were: (a) diagnosis of a

severe mental disorder (e.g., psychosis, substance abuse or dependence, eating or personality disorders); (b) frequent or recent suicide attempt(s); and (c) individuals receiving psychological treatment for any other mental disorder. In addition, after treatment, the following question was included in the questionnaire battery to determine whether the participants were following another therapy in addition to the one established in the study: "*Since the beginning of your participation in this study, have you received any other psychological or psychiatric therapy (public or private)*?". Participants had to answer yes or no. If the answer was yes, the participant was automatically excluded from the study. The sociodemographic and clinical characteristics of the sample are shown in Table 1.

## Ethical aspects

The study was conducted in accordance with the Declaration of Helsinki. This project is a multi-center Randomized Clinical Trial with medication (N EUDRACT: 2013-001955-11; protocol code: ISRCTN58437086) promoted by the Psicofundación (The Spanish Foundation

**Table 1. Sociodemographic and clinical characteristics of participants at baseline.**

|  | Participants *N* = 1061 |
|---|---|
| *Demographic characteristics* | |
| Age, mean (*SD*) | 43.0 (11.8) |
| Sex: women, *n* (%) | 861 (81.1) |
| Marital status, *n* (%) | |
| Married | 513 (48.4) |
| Divorced | 87 (8.2) |
| Widowed | 29 (2.7) |
| Separate | 58 (5.5) |
| Never Married | 212 (20.0) |
| Unmarried | 162 (15.3) |
| Level of Education, *n* (%) | |
| No schooling | 11 (1.0) |
| Basic education | 267 (25.2) |
| Secondary education | 233 (22.0) |
| High School | 262 (24.7) |
| Bachelor | 242 (22.8) |
| Master/Doctorate | 46 (4.3) |
| Employment situation, *n* (%) | |
| Part-time employee | 392 (36.9) |
| Employed full time | 180 (17.0) |
| Unemployed, in search of work | 230 (21.7) |
| Unemployed, not looking for work | 137 (12.9) |
| Temporary incapacity to work | 73 (6.9) |
| Permanent incapacity to work | 23 (2.2) |
| Retired | 26 (2.5) |
| *Clinical characteristics* | |
| Symptoms, mean (*SD*) | |
| PHQ-9 | 13.6 (5.4) |
| GAD-7 | 12.3 (4.6) |

*Note.* SD = Standard deviation; GAD-7 = Generalized Anxiety Disorders-7; PHQ-9 = Patient Health Questionnaire-9 (depression).

for the Promotion of the Scientific and Professional Development of Psychology). The research was approved by the CEIC-APCV—the national research ethics committee coordinator—and the Spanish Medicines and Health Products Agency. Approval was received by both agencies in November 2013, prior to study initiation in December 2013.

Only direct members of the internal study team can access the data. A more detailed description of data confidentiality can be found in the study protocol [42].

## Sample

A total of 1061 participants were randomised to receive TAU alone (n = 527) or combined treatment involving TD-GCBT + TAU (n = 534).

## Measures

The following assessment measures were administered at five different time points (pre-treatment, post-treatment and 3-, 6- and 12-month follow-up).

*Patient Health Questionnaire-9 (PHQ-9)* [44]. The PHQ-9 is a screening instrument that is widely used in PC to assess the frequency of depressive symptoms over the last 2 weeks [46]. The instrument consists of nine items scored from 0 (*not at all*) to 3 (*nearly every day*) with a maximum score of 27. A score of ten or higher is considered a good cut-off point for establishing the presence of depression. This instrument has been shown to offer good internal consistency in a Spanish population (α = 0.82) [46] and also in the clinical trial conducted by Cano-Vindel [36] (α = 0.86).

*Generalized Anxiety Disorder-7 (GAD-7)* [45]. The GAD-7 is a screening instrument widely used in PC to assess the frequency of anxiety symptoms over the last 2 weeks [47]. The instrument consists of 7 items scored from 0 (*not at all*) to 3 (*nearly every day*) with a maximum score of 21. A score of ten or higher is considered a good cut-off point for establishing the presence of anxiety. This instrument has been validated in a Spanish population [48] and has been shown to offer good internal consistency in the clinical trial conducted by Cano-Vindel [36] (α = 0.87).

## Description of treatments

**Treatment as usual (control group).** The control group included participants receiving TAU, also described as standard or pharmacological treatment [36]. TAU consisted of routine consultations with the GPs in face-to-face sessions (5–7 minutes) to assess the patients' physical and/or psychological complaints and included the prescription of antidepressants, anxiolytics or hypnotics, and/or informal counselling/support.

**Transdiagnostic group cognitive behavioural therapy (experimental group).** The experimental group included patients receiving transdiagnostic group cognitive behavioural therapy (TD-GCBT) plus treatment as usual (TAU). Patients receiving pharmacological treatment prior to the start of the study could also be randomly assigned to the experimental group. Once assigned, primary care physicians were not allowed to prescribe new medications or increase pharmacotherapy to these participants, but could decrease or eliminate medication if there was improvement. The treatment consisted of seven sessions of 90 minutes each held over a period of approximately 12–14 weeks in small groups (8–10 patients) at the PC centre. The sessions were conducted by clinical psychologists who had been previously trained in the treatment protocol through an 8-hour training programme led by a senior clinical psychologist (for a detailed description see [36]).

The TD-GCBT protocol modules are as follows: *(1) introduction* and *psychoeducation*: presentation and explanation of the protocol and information to participants about emotions,

their adaptive function and when they become maladaptive and turn into EDs; *(2) relaxation*: reduction of psychophysiological activation through different self-regulation strategies (diaphragmatic breathing, progressive muscle relaxation and visualization); *(3) Cognitive restructuring*: information about rational and irrational thoughts and strategies to modify them; *(4) behavioral therapy*: behavioral activation, exposure techniques, social skills and problem solving and *(5) relapse prevention*: acceptance of relapses and restructuring of relapses. (for a detailed description see [36]).

## Statistical analyses

All analyses were performed in R Studio (version 4. 2. 2). Our study is a secondary analysis of a randomized clinical trial where an intention-to-treat analysis was performed including all randomized patients using the chained equation multiple imputation procedure with five imputations [36]. The evaluations were completed by the following number of participants: At PRE measure (TAU = 534; TD-GCBT + TAU = 527), at POST (TAU = 316; TD-GCBT + TAU = 315), at 3-month follow-up (TAU = 238; TD-GCBT + TAU = 273), at 6-month follow-up (TAU = 205; TD-GCBT + TAU = 229), and at 12-month follow-up (TAU = 180; TD-GCBT + TAU = 208) [36].

## NIA (Network Intervention Analysis)

NIA [33] was used to investigate the direct and differential effects produced by treatment on specific symptoms and the impact of interventions on the network structure over time. In other words, NIA identifies whether some of the symptoms included in the network are more strongly affected by one of the treatments (i.e., direct treatment-specific effects). A mixed graphical model (MGM) was applied to estimate a network for each assessment point (pre-treatment, post-treatment and 3-, 6- and 12-month follow-up) using the R package *mgm* (version 1.2–12) [49]. This model is useful to represent complex systems and to obtain information about the relationship between variables of different types, including binary, ordinal, categorical and continuous variables, among others. In addition, MGM allows representing the interaction between two nodes after controlling for associations with all the other variables of the network [50]. In the present study, the networks included depressive and anxious symptoms as continuous variables and treatment allocation as a binary variable (0 = TAU, 1 = TD-GCBT + TAU). Graphical LASSO (least absolute shrinkage and selection operator) was used to regularize models and reduce potential spurious edges [51]. We use the cross-validation approach to select the adjustment parameter and specify the degree of regularization. Specifically, we estimated the networks using a gamma hyperparameter of 0.25 to control the amount of regularisation applied.

To visualise between-group differences in symptom severity, item-means were standardised to baseline for each group separately using means and standard deviations. Differences between the TAU group and the TD-GCBT + TAU group were then compared. Negative values indicate a greater decrease in symptom severity in the experimental group compared to the control group (see Table 2). In the network, the results reflect the size of the nodes: a smaller size represents a larger effect of the TD-GCBT + TAU intervention. In this way, we can observe treatment-induced changes over time.

To evaluate the accuracy of the edge-weights in the networks, we used the *resample* () function implemented in the *mgm* package (number of bootstrap samples). For each network, we ran 1000 bootstrap samples for which we fitted the model and plotted the resulting sampling distribution of all edges using the function *plotRes* () of the *mgm* package. The plot (see S1–S5 Figs) shows the 5% and 95% quantiles of the sampling distribution [33].

**Table 2. Group differences in changes in PHQ-9 and GAD-7 item severity over time.**

|  | Anh | SadMood | Sleep | Energy | Appet | Guilt | Concen | Mot | Sui | Nerv | ConWorry | TMWorry | Relax | Rest | Irri | Afra |
|---|---|---|---|---|---|---|---|---|---|---|---|---|---|---|---|---|
| Baseline | -0.02 | 0.02 | 0.03 | 0.08 | 0.00 | 0.10 | 0.01 | 0.00 | -0.00 | 0.15 | 0.06 | 0.04 | 0.06 | -0.01 | 0.03 | -0.00 |
| Post-treatment | -0.56 | -0.72 | -0.52 | -0.59 | -0.44 | -0.50 | -0.45 | -0.43 | -0.36 | -0.64 | -0.64 | -0.75 | -0.81 | -0.58 | -0.56 | -0.43 |
| Three months | -0.31 | -0.42 | -0.35 | -0.26 | -0.24 | -0.22 | -0.35 | -0.34 | -0.16 | -0.28 | -0.30 | -0.37 | -0.50 | -0.39 | -0.25 | -0.18 |
| Six months | -0.42 | -0.38 | -0.33 | -0.41 | -0.23 | -0.19 | -0.28 | -0.35 | -0.15 | -0.40 | -0.44 | -0.43 | -0.40 | -0.32 | -0.24 | -0.27 |
| Twelve months | -0.25 | -0.31 | -0.43 | -0.55 | -0.27 | -0.22 | -0.37 | -0.19 | -0.13 | -0.49 | -0.50 | -0.45 | -0.54 | -0.32 | -0.38 | -0.28 |

*Note*. Standardized change scores were computed at each assessment points. Differences in changes between the two groups were then calculated. **Values** indicate the magnitude of difference between the groups, rather than the magnitude of change for any individual group. A value of 0 indicates that the TAU alone and TD-GCBT + TAU group exhibited the same degree of change. **Positive** values indicate that the TAU alone group reported larger declines in symptom severity from baseline than the TD-GCBT + TAU group. **Negative** values indicate that the TD-GCBT + TAU group reported larger declines in symptom severity from baseline than the TAU alone group. Anh = anhedonia. SadMood = sad mood. Sleep = trouble sleeping. Energy = low energy. Appet = Appetite change. Guilt = Feeling of worthlessness. Concen = concentration difficulties. Mot = psychomotor agitation/retardation. Sui = thoughts of death. Nerv = nervousness or anxiety. ConWorry = uncontrollable worry. TMWorry = worry too much. Relax = trouble relaxing. Rest = restlessness. Irri = irritable. Afra = afraid something will happen.

## Results

Fig 1 shows the five estimated networks for each assessment point (see also S6–S10 Figs). The circular nodes represent the PHQ-9 and GAD-7 symptoms, and the square node indicates the treatment condition (TAU or TD-GCBT + TAU). A link between two nodes represents the only association between two variables, after controlling for all other variables in the network. The thickness of the links is proportional to the relative strength of the association. The largest and most consistent positive associations (green links) over time were found between the items *interest or pleasure* (D1) and *feeling down, depressed or hopeless* (D2), as well as between *not being able to stop or control worrying* (A2) and *worrying too much about different things* (A3).

Fig 1 shows a direct link at baseline between TAU (green link) and the symptoms *feeling nervous, anxious or on edge*, while the transdiagnostic group CBT shows the greatest impact at all other time points. Specifically, the results revealed direct effects at post-treatment of TD-GCBT + TAU on the anxiety symptoms *worrying too much about different things* (A3), *trouble relaxing* (A4), *being so restless that it is hard to sit still* (A5), *becoming easily annoyed or irritable* (A6) and the depressive symptoms *feeling down, depressed or hopeless* (D2). In subsequent assessments up to the one-year follow-up, differences between treatments continued to show a greater effect of TD-GCBT + TAU on certain specific symptoms compared to TAU. Specifically, at 3 month follow-up, TD-GCBT + TAU was associated with a decrease in *difficulty relaxing* (A4), *restlessness* (A5), *concentration problems* (D7) and *sad mood* (D2). Subsequently, at 6-month follow-up, differences were observed in favour of the treatment received by the experimental group in *irritability* (A6), *psychomotor problems* (D8) and *anhedonia* (D1). Finally, at 12 month follow-up, TD-GCBT + TAU was particularly efficacious for the symptoms *difficulty relaxing* (A4) and *increased energy* (D4).

## Discussion

The aim of the present study was to establish whether two treatments (TD-GCBT + TAU vs. TAU alone) for EDs could influence symptoms differently over time. For this purpose, we applied NIA and estimated five network models (pre-treatment, post-treatment and 3-, 6- and 12-month follow-up). We expected each treatment to show a different pathway of action.

The main finding of our study was the direct association between TD-GCBT + TAU and some specific symptoms of anxiety and depression, whereas TAU was not directly associated with any symptoms differentially, indicating favourable effects of TD-GCBT + TAU compared

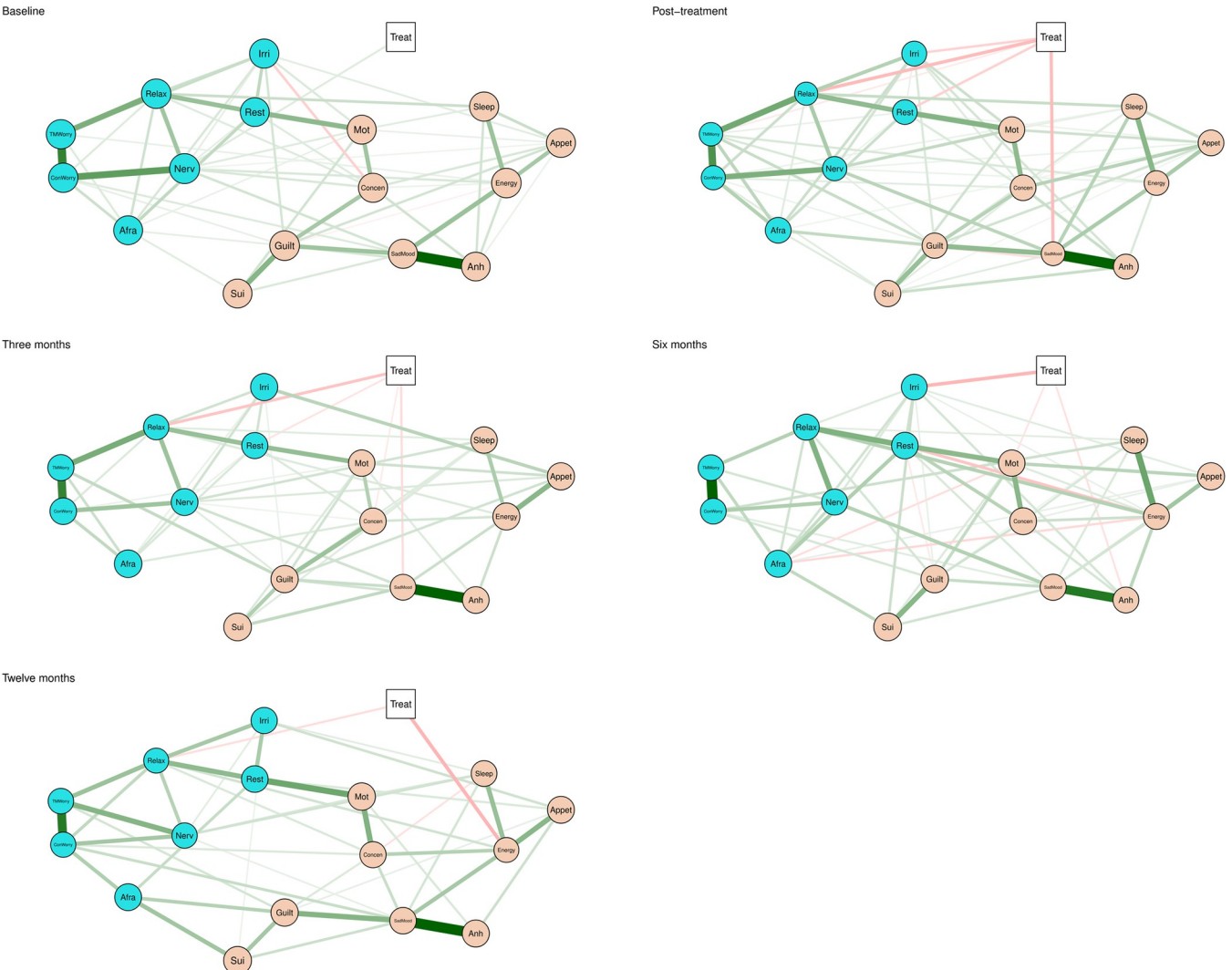

**Fig 1. Estimated networks pre-treatment (baseline), post-treatment and 3-, 6- and 12-month follow-up.** The nodes in the network represent the seven items of the Generalized Anxiety Disorder-7 and the nine items of the Patient Health Questionnaire-9 (circles), and treatment (square). The edges represent the unique association between two variables, after conditioning on all other variables in the network. Green edges represent positive associations, red edges represent negative associations, and the width and color-saturation of the edges are proportionate to the strength of the association and can be compared across networks. Associations between the treatment variable and a symptom indicate that symptom to be more directly affected by only one of the treatments. Green edges indicate a positive treatment effect for TD-GCBT + TAU, and red edges indicate a positive treatment effect for TAU alone. Abbreviations: Treat = treatment. Anh = anhedonia. SadMood = sad mood. Sleep = trouble sleeping. Energy = low energy. Appet = Appetite change. Guilt = Feeling of worthlessness. Concen = concentration difficulties. Mot = psychomotor agitation/retardation. Sui = thoughts of death. Nerv = nervousness or anxiety. ConWorry = uncontrollable worry. TMWorry = worry too much. Relax = trouble relaxing. Rest = restlessness. Irri = irritable. Afra = afraid something will happen.

to TAU alone. These results are similar to previous traditional research that has largely demonstrated a greater effect of TD-CBT in reducing clinical symptoms compared to TAU [26, 27, 36], thus suggesting that the addition of TD-GCBT to TAU is likely to exert a beneficial influence on clinical symptom improvement in patients with EDs. Despite this, other research has highlighted that the use of psychotropic drugs, such as benzodiazepines, may interfere negatively with some learning processes such as habituation [52]. In addition, scientific evidence advises against the administration of anxiolytics and antidepressants in most cases because of their side effects and their limited long-term effect [53].

Our work incorporates a novel approach underpinned by network methodology and provides additional and specific information at the symptom level (specific associations between network elements) that are directly and differentially affected by either of the treatment conditions. Specifically, our results revealed that TD-GCBT + TAU was directly associated with four specific anxiety symptoms (*worrying too much about different things*, *difficulty relaxing*, *restlessness* and *irritability*) and one depression symptom (*sad mood*), showing greater differential effects than TAU in symptom reduction after treatment (see Table 2). These results support those of the main study of Cano-Vindel [36] who observed significant differences in favour of TD-GCBT + TAU in the overall reduction of anxiety and depressive symptoms using a sum score. Therefore, it seems possible that TD-GCBT can effectively modify, both directly and indirectly, the symptoms targeted by the different components included in the protocol modules (psychoeducation, relaxation, cognitive restructuring techniques, behavioural therapy and relapse prevention) [42]. Thus, the implementation of NIA is useful for revealing specific treatment pathways of action, which may favour the development of more effective interventions to address the symptoms of emotional disorders. In our case, it is likely that the addition of TD-GCBT may help patients to decrease their physiological arousal and tolerate physical and emotional sensations (*difficulty relaxing*, *restlessness* and *irritability*), and to develop cognitive reappraisal strategies useful for changing the way an emotion is experienced and to generate more realistic interpretations (*sad mood* and *worrying too much about different things*) [54].

Furthermore, our results indicate that the impact of TD-GCBT on specific symptoms is maintained in the long term (one year in this case). Specifically, direct associations were observed on specific anxiety symptoms (e.g., *difficulty relaxing*, *restlessness* and *irritability*), just as new differential effects on depressive symptoms (*anhedonia*, *sad mood*, *concentration problems*, *psychomotor problems* and *energy*) gradually emerged. Based on our results, we hypothesized that TD-GCBT, compared to TAU, might be contributing to a greater extent to the improvement of two key symptoms of EDs (*sad mood* and *anhedonia*). Afterwards, TD-GCBT could increase energy status by improving *sad mood* and *anhedonia*. Although further research is needed, it is possible that these changes are favoured by psychological treatment (e.g., through the effects exerted by behavioral activation). These results support the network analysis proposition that treatments appear to affect specific symptoms first and subsequently trigger a wave of changes in other symptoms indirectly, thus modifying the connections between network elements. That is, an intervention may promote changes in certain variables that, in turn, lead to changes in others. This is consistent with the phenomenon known as *hysteresis*, or the process of activation between symptoms, even after the triggering cause has disappeared [37].

On the other hand, it should be noted that there is temporal variability in the direct effects of TD-GCBT on some symptoms. For example, direct associations on irritability emerge at post-treatment and at 6 months, but not at 3 or 12 months follow-up. Similar results were observed in the difficulty to relax. Several reasons could explain such variability in the effects of TD-GCBT: first, the order established in the treatment modules; second, the variability between subjects in the training and in the regular practice followed by each participant after the end of the treatment and, finally, a statistical explanation based on the Regularization process (i.e., if there is a similar effect at two time points, the link between two nodes will appear exclusively in the one that presents statistical significance). However, this does not mean that there are no effects on these symptoms at the other time point, but that these have not been significant compared to the other time point.

However, more research is needed to determine the direct and indirect effects of treatments on mental disorders in general and EDs in particular, which could help to improve the effectiveness of treatments through symptom-targeted interventions. In this regard, it would be

interesting to analyse other trials with transdiagnostic treatments or other effective therapies that can be implemented in the context of PC, such as brief therapies, and to examine changes at the symptom level. On the other hand, future research could incorporate more assessments over time to examine long-term symptom improvement. In addition, it may be of interest to specifically analyse how each module of the protocol affects each symptom, and to observe whether these changes are sustained over time, which would help to determine which intervention strategies would be more likely to improve particular symptoms (e.g., people who regularly practise relaxation techniques may be more likely to improve their level of physiological arousal). Importantly, as Mullarkey pointed out [40], these differential treatment effects are masked when sum scores are used to examine changes in symptom severity, making the application of the NIA relevant and providing information that complements these obstacles reported in previous research.

Two of the symptoms directly affected in our study, *worrying too much* and *sad mood*, have been identified as core symptoms through network analysis in previous research [55, 56]. From network theory, it is proposed that intervening directly and effectively on these core symptoms could be a promising strategy that could lead to a cascading decrease in other symptoms [57]. In this sense, TD-GCBT + TAU directly decreased the symptom *worrying too much*, and in turn, significantly decreased the symptoms (*uncontrollable worries*, *nervousness* or *anxiety* and *difficulty relaxing*) strongly associated with this core symptom. On the other hand, transdiagnostic therapy directly reduced *sad mood*, which is strongly associated with other symptoms (*anhedonia*, *low energy*, or *guilt*) that improved significantly throughout the treatment and in the months following the end of treatment. However, as noted above, it would be appropriate to develop other designs that include more assessments during treatment to determine more solidly the trajectory of symptom changes produced by transdiagnostic intervention or other treatments for emotional disorders. Or, on the other hand, studies incorporating centrality measures could be considered to design specific interventions aimed at experimentally manipulating these nodes and observing changes in the network.

In terms of clinical implications, our results highlight the importance of understanding the potential pathways through which treatments exert their effects. In our study, both the absence of differential effects of TAU on specific symptoms and the direct effects of TD-GCBT + TAU on specific symptoms of anxiety and depression support the results that highlight the beneficial effects of adding TD-GCBT to treat mild-moderate EDs in primary care settings. Defining these specific underlying treatment pathways of action could be useful to improve treatment efficacy (e.g., by combining different interventions or applying specific techniques to modify certain variables). These findings are in line with the need to clarify the key mechanisms on which we could rely to design or adjust interventions for specific psychological problems in a targeted way [35].

Some limitations of our study need to be mentioned. First, we only examined effects after treatment. We encourage future studies to include more measures before and during the intervention that would allow us to identify more comprehensively the order of TD-GCBT-induced effects on anxiety and depressive symptoms throughout treatment. Second, only symptomatic variables were included in the network model and future research would benefit from including other variables that are clinically relevant for EDs (e.g., neuroticism or emotional regulation strategies). Third, although the sample size is large, the presence of stronger direct effects may have been affected by the variation in the number of participants at different time points. Fourth, it is unlikely that the symptomatic variables of EDs selected in this study perfectly capture all variables relevant to understanding the effect of TAU or TD-GCBT + TAU on specific symptoms. Thus, future research should take into account the comorbidity between anxiety, depression, somatizations, and panic and include other variables of interest. Fifth, the findings

of the present study can only be considered for mild and moderate symptoms. Sixth, since this is a very specific sample, the results cannot be generalized to other populations. Finally, future studies are needed to analyse whether there are differences between individual and group format interventions.

In conclusion, the current study shows the viability and utility of applying this novel network methodology (NIA) and highlights the importance of examining treatment-specific pathways and the usefulness of NIA in providing complementary information to overcome the limitations of previous research. Thus, identifying the specific effects of interventions on symptoms could help to select the optimal treatment based on the symptoms a person presents with, thus opening the door to the development of more effective treatments. In other words, knowing how the effects of the intervention develop could help to reorganise the treatment modules and select the techniques that are best suited to patients' needs. Therefore, we believe that the new NIA approach can be a useful tool for examining the effects of treatments at the symptom level at different time points.

## Supporting information

**S1 Fig. Bootstrapped sampling distribution of edge weight estimates at pre-treatment.**
(PDF)

**S2 Fig. Bootstrapped sampling distribution of edge weight estimates at post-treatment.**
(PDF)

**S3 Fig. Bootstrapped sampling distribution of edge weight estimates at 3-month follow-up.**
(PDF)

**S4 Fig. Bootstrapped sampling distribution of edge weight estimates at 6-month follow-up.**
(PDF)

**S5 Fig. Bootstrapped sampling distribution of edge weight estimates at 12-month follow-up.**
(PDF)

**S6 Fig. Regularized network at pre-treatment.**
(PDF)

**S7 Fig. Regularized network at post-treatment.**
(PDF)

**S8 Fig. Regularized network at three months follow-up.**
(PDF)

**S9 Fig. Regularized network at six months follow-up.**
(PDF)

**S10 Fig. Regularized network at twelve months follow-up.**
(PDF)

## Acknowledgments

The authors wish thank to all the patients who participated in the study and to all members of PsicAP-Cost project.

## Author Contributions

**Conceptualization:** Francisco Jurado-González, Francisco García-Torres, Alba Contreras, Juan A. Moriana.

**Data curation:** Roger Muñoz-Navarro, Leonardo Adrián Medrano, Paloma Ruiz-Rodríguez, Antonio Cano-Vindel.

**Formal analysis:** Francisco Jurado-González, Alba Contreras.

**Funding acquisition:** Antonio Cano-Vindel, Juan A. Moriana.

**Investigation:** Francisco Jurado-González, Francisco García-Torres, Juan A. Moriana.

**Methodology:** Francisco Jurado-González, Francisco García-Torres, Alba Contreras, Juan A. Moriana.

**Project administration:** César González-Blanch, Antonio Cano-Vindel, Juan A. Moriana.

**Resources:** Antonio Cano-Vindel, Juan A. Moriana.

**Software:** Francisco Jurado-González, Alba Contreras.

**Validation:** Francisco García-Torres, Alba Contreras, Roger Muñoz-Navarro, César González-Blanch, Leonardo Adrián Medrano, Paloma Ruiz-Rodríguez, Eliana M. Moreno, Carolina Pérez-Dueñas, Antonio Cano-Vindel, Juan A. Moriana.

**Visualization:** Francisco Jurado-González.

**Writing – original draft:** Francisco Jurado-González.

**Writing – review & editing:** Francisco García-Torres, Alba Contreras, Roger Muñoz-Navarro, César González-Blanch, Leonardo Adrián Medrano, Paloma Ruiz-Rodríguez, Eliana M. Moreno, Carolina Pérez-Dueñas, Antonio Cano-Vindel, Juan A. Moriana.

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
