## [Decision Letter · Decision Letter 0]

8 Nov 2023

PONE-D-23-20048Comparing psychological versus pharmacological treatment in emotional disorders: A network analysisPLOS ONE

Dear Dr. González,

Thank you for submitting your manuscript to PLOS ONE. After careful consideration, we feel that it has merit but does not fully meet PLOS ONE’s publication criteria as it currently stands. Therefore, we invite you to submit a revised version of the manuscript that addresses the points raised during the review process.

We look forward to receiving your revised manuscript.

Kind regards,

Chiedu Eseadi, PhD

Academic Editor

PLOS ONE

Journal Requirements:

2.In your Data Availability statement, you have not specified where the minimal data set underlying the results described in your manuscript can be found. PLOS defines a study's minimal data set as the underlying data used to reach the conclusions drawn in the manuscript and any additional data required to replicate the reported study findings in their entirety. All PLOS journals require that the minimal data set be made fully available. For more information about our data policy, please see http://journals.plos.org/plosone/s/data-availability.

Reviewers' comments:

Reviewer's Responses to Questions

**Comments to the Author**

1. Is the manuscript technically sound, and do the data support the conclusions?

Reviewer #1: Yes

Reviewer #2: Yes

Reviewer #3: Yes

2. Has the statistical analysis been performed appropriately and rigorously? 

Reviewer #1: Yes

Reviewer #2: Yes

Reviewer #3: Yes

3. Have the authors made all data underlying the findings in their manuscript fully available?

Reviewer #1: Yes

Reviewer #2: Yes

Reviewer #3: Yes

4. Is the manuscript presented in an intelligible fashion and written in standard English?

Reviewer #1: Yes

Reviewer #2: Yes

Reviewer #3: Yes

5. Review Comments to the Author

Reviewer #1: As the statistical reviewer I will focus on methods and reporting.

1) the authors state "...reduce potential spurious edges by selecting the tuning parameter called ‘cross-validation’". I struggle to understand what they mean, especially in regarding cross-validation, which is a technique and not a parameter. Can they clarify and rephrase please.

2) clarify what the resample() function does, as it is currently unclear (is it the bootstrap)

3) clarify how time is modelled into the analysis. was it a separate model for each time point?

4) clarify all the covariates used in the analyses, as per the STROBE statement, in the methods section.

5) clarify what was done with missing data, levels of missingness (per variable and overall dropped in the model). if a large number of observations is dropped, why wasn't multiple imputation used?

Reviewer #2: This article sheds some light on how a group transdiagnostic treatment has a greater effect on certain symptoms of anxiety and depression compared to pharmacological treatment in primary care through a network analysis. It therefore has a novel methodology that can provide directions for new therapeutic approaches and individualization of treatments. However, I have some doubts and comments that may improve the quality of the article.

1. The authors talk both in the introduction and in the discussion about “mechanisms of change” to refer to the contributions of their work. However, a mechanism of change attempts to explain why a treatment works and this study focuses on identifying the sequence of symptoms most affected. I consider that the terminology used should be modified and perhaps opt for others such as “path of action”, which is also used at some point in the manuscript.

2. Diagnostic interview is said to be used to identify major depressive disorder. Why is it not applied to generalized anxiety disorder?

3. In the original clinical trial, there are more measures of symptoms, such as measures for somatoform disorder or panic disorder. Why are these measures not included in this study? Somatic problems are mentioned within emotional disorders, but they are not analyzed later.

4. In the discussion it is said “These results are similar to previous traditional research that has largely demonstrated a greater effect of TD-CBT in reducing clinical symptoms compared to TAU [24,25,34], thus suggesting that the addition of TD-CBT GCBT to TAU is likely to exert a beneficial influence on clinical symptom improvement in patients with EDs.” I agree, but not only would it not add anything, but it could potentially interfere with the therapeutic process (there is evidence that reducing anxious symptoms through drugs can reduce the habituation response to anxiety disorders). This is without counting the costs and side effects. Perhaps the authors will consider expanding this argument.

5. Also, in the discussion it is said that the treatment would be beneficial to increase emotional regulation. Although I believe that this statement may be true, the results of this study do not fully support this, since of the core symptoms, two of them have more to do with the physiological state (relaxation and rest) than with a cognitive process.

6. According to the results, it would seem that the most immediate benefit of transdiagnostic group therapy would be in anxiety symptoms and, subsequently, in depressive symptoms. It is argued that it could be the former who bring about change in the latter. In that case, would it be wise to focus exclusively on a treatment for anxiety disorders and hope that it would also have an effect on depressive symptoms? What would justify the use of a transdiagnostic approach then? Or is it the treatment itself that takes time to take effect? Would the intervention change depending on the core disorder or symptoms? It would be helpful for the authors to reflect on these issues and outline how treatments can be individualized to be useful to clinicians.

7. In the limitations section, it would be convenient to add that these results are for mild and moderate symptoms, since they could change substantially when dealing with serious emotional disorders. Furthermore, the extent to which comorbidity between anxiety, depression, somatization and panic can influence the results is not studied. Finally, it would also be appropriate to comment that there could be differences between the group and individual treatment formats.

Reviewer #3: This study employs an innovative approach by utilizing network analysis methodology to investigate the dirrect and differential effects of TD-GCBT+TAU and TAU on symptoms of anxiety and depression. The findings demonstrate a direct association between TD-CBT+TAU and different anxiety and depression symptoms over time, highlighting a more pronounced effect of TD-CBT+TAU in reducing emotional symptoms compared to TAU alone. This study contributes significantly to our understanding of the functioning of CBT and underscores its relevance in the context of primary care. The manuscript has several notable strengths, including its extensive sample size and the inclusion of multiple follow-up time points, which enhance the robustness of the study. From a general point of view I consider that the study is well written but I have some minor comments:

1. In the first paragraph of the introduction, the author mentions emotional disorders, highlighting depression, anxiety, and other related disorders. It would be beneficial for the author to specify the scope and definition of these categories to provide clarity to the readers. Defining the specific disorders encompassed within the term "other related disorders" will enhance the overall understanding of the study's context.

2. Within the same paragraph, the manuscript presents statistics on the prevalence of depressive and anxiety disorders, citing data from a study conducted in 2017. Given that the manuscript also alludes to the potential exacerbation of these effects following the pandemic, it would be advisable to update these epidemiological figures. The COVID-19 pandemic has had a significant impact on mental health, and it is likely that the prevalence of these disorders has increased since 2017. Updating this information with more recent data would provide a more accurate and relevant context for the study.

3. While the Measures section delves into detailed descriptions of the instruments used in the study, it would be beneficial to specify this information in the Study Design and Procedure section as well. In the current text, there is a lack of detailed information regarding the questionnaires employed, with only a brief mention of the PHQ (Patient Health Questionnaire).

4. In the manuscript, it is mentioned that "Patients with PHQ scores indicative of major depression were re-evaluated by a clinical psychologist using a semi-structured interview". It would be interesting to clarify the specific cut-off point employed to determine the criteria for major depression. Furthermore, it would be beneficial to understand the basis of this interview and the specific criteria used for the inclusion or exclusion of participants in the study.

5. It is my understanding that the presence of a major depressive disorder is considered an exclusion criterion in the study. However, the inclusion and exclusion criteria do not explicitly specify this information. Instead, they refer to the "presence of emotional symptomatology" without detailing the severity of these symptoms.

6. Furthermore, in the Methods section, it is mentioned that "patients presented signs or symptoms of negative emotional problems, moderate depression, anxiety, or somatic symptoms." However, in the Discussion, reference is made to "mild-moderate EDs." This inconsistency in terminology could lead to confusion among readers regarding the study's eligibility criteria.

7. Regarding the measures used in the study, it would be valuable to specify whether these instruments are self-administered by the participants.

8. I recommend verifying the data presented in Table 1. Specifically, in the "Marital Status" category, it appears that the number of individuals exceeds the sample size specified in the study.

9. My understanding is that the experimental group combines TD-GCBT and TAU alone. It would be beneficial to provide details on how these treatments are combined. Is there a specific protocol with a defined number of TAU sessions? Can these treatments overlap in time?

10. Furthermore, it would be valuable to provide a concise description of the TD-GCBT protocol. The manuscript makes references to different modules in the discussion, but these modules are not explained in detail earlier in the text. Providing a brief overview of the treatment protocol will help readers understand the specific components and methodologies involved in TD-CBT, which is crucial for interpreting the study's findings and relevance.

11. During the various assessment time points, has there been any monitoring or control to ascertain whether patients engaged in other forms of psychological therapy beyond the established sessions? While I understand that not receiving any other psychological therapy during the established sessions is an exclusion criterion, it is not clear whether patients are permitted to pursue additional therapy during the follow-up period.

12. In the Results section, there is a sentence that reads as follows: "Specifically, at follow-up, TD-GCBT+TAU was associated with a decreased..." My concern pertains to the term "follow-up." While it appears that this term may refer to post-treatment based on the data presented in the figure, it would be beneficial to explicitly clarify which specific measure of follow-up is being referenced. Ensuring this clarification will eliminate potential ambiguity and enhance the readers' understanding of your results.

13. I have noted that in the Discussion section, you mention the variation in the number of participants at different time points as a limitation of the study. I would like to obtain more information on this matter. Could you provide details on how many participants were considered at each assessment time point? Were any data imputation methods employed to address potential variations in sample size at different time points?

14. It would be interesting to specify in the discussion that the results cannot be generalized to other populations, as the one considered in the study is very specific.

15. The authors stated that "treatments appear to affect specific symptoms first and subsequently trigger a wave of changes in other symptoms indirectly, thus modifying the connections between network elements." However, how does the author explain the occurrence of these changes in variables like anhedonia or energy, but not in other variables such as sleep and appetite?

16. Similarly, there is an observed loss of association in variables like irritability at three and twelve months, but not at six months. A similar pattern is also noted with difficulty in relaxing. How can this variation be explained?

6. PLOS authors have the option to publish the peer review history of their article (what does this mean?). If published, this will include your full peer review and any attached files.

Reviewer #1: No

Reviewer #2: No

Reviewer #3: No

---

## [Author Response · Author response to Decision Letter 0]

11 Jan 2024

Emily Chenette. Editor in Chief of PLOS ONE

Dear Dr. Emily Chenette,

 We would like to thank you and the reviewers for carefully reviewing our paper entitled “Comparing psychological versus pharmacological treatment in emotional disorders: A network analysis”. Reference PONE-D-23-20048. We have made changes following the valuable suggestions, what we believe have strengthened the manuscript, helping us to improve our manuscript. 

REVIEWER #1: 

Comment #1: The authors state "...reduce potential spurious edges by selecting the tuning parameter called ‘cross-validation’". I struggle to understand what they mean, especially in regarding cross-validation, which is a technique and not a parameter. Can they clarify and rephrase please.

Response: We have been able to re-examine and reformulate the sentence, as follows: “We use the cross-validation approach to select the adjustment parameter and specify the degree of regularization ”. (p. 11-12, para. 2). 

Comment #2: Clarify what the resample() function does, as it is currently unclear (is it the bootstrap).

Response: The resample () function implemented in the mgm package allows to evaluate the accuracy of the edges weights (resampling number). In other words, it allows to estimate the accuracy of the networks by using subsets of the available data (i.e. sampling with replacement of the original sample). (p. 14, para. 1).

Comment #3: Clarify how time is modelled into the analysis. was it a separate model for each time point?

Response: Indeed, a different model was estimated for each time point (pre-treatment, post-treatment and 3-, 6- and 12-month follow-up), as described in the "statistical analyses" section. (p. 11, para. 2). 

Comment #4: Clarify all the covariates used in the analyses, as per the STROBE statement, in the methods section.

Response: We thank the reviewer for the suggestion to clarify the covariates used according to the STROBE statement. In our study, covariates, i.e., those variables that may influence the dependent variable and may be related to the independent variable, were not taken into account. However, we only consider the independent variable (treatment) as a covariate for the purposes of statistical analysis.

Comment #5: Clarify what was done with missing data, levels of missingness (per variable and overall dropped in the model). if a large number of observations is dropped, why wasn't multiple imputation used?

Response: Following the reviewer's recommendations, we have clarified that our study is a secondary analysis of a randomized clinical trial where an intention-to-treat analysis was performed, including all randomized patients using the chained equations multiple imputation procedure, with five imputations. (p.11, para. 1).

REVIEWER #2:

Comment #1: The authors talk both in the introduction and in the discussion about “mechanisms of change” to refer to the contributions of their work. However, a mechanism of change attempts to explain why a treatment works and this study focuses on identifying the sequence of symptoms most affected. I consider that the terminology used should be modified and perhaps opt for others such as “path of action”, which is also used at some point in the manuscript.

Response: Indeed, the concept "pathways of action" is more appropriate for the purpose of our study (to identify the sequence of symptoms most affected by treatment). Therefore, we have eliminated the concept "mechanisms of change" and, instead, we have written "pathways of action", thus unifying the terminology throughout the text.

Comment #2: Diagnostic interview is said to be used to identify major depressive disorder. Why is it not applied to generalized anxiety disorder?

Response: Severe anxiety disorders were a reason for exclusion from the study, and were therefore also identified through the diagnostic interview with the clinical psychologist. We have made the relevant clarifications in the "Study design and procedure" section. (p. 7, para. 1).

Comment #3: In the original clinical trial, there are more measures of symptoms, such as measures for somatoform disorder or panic disorder. Why are these measures not included in this study? Somatic problems are mentioned within emotional disorders, but they are not analyzed later.

Response: We also considered the possibility of including somatic symptoms in network analyses. However, in complex systems research, it is advisable to start from simpler models and then add other components (e.g. somatic symptoms) that allow progressing towards more complex models that deepen the understanding of the system. Consequently, we believe and consider it necessary that future research should incorporate somatic symptoms in the analysis in order to understand more precisely the effect of treatments on emotional disorders. 

Comment #4: In the discussion it is said “These results are similar to previous traditional research that has largely demonstrated a greater effect of TD-CBT in reducing clinical symptoms compared to TAU [24,25,34], thus suggesting that the addition of TD-CBT GCBT to TAU is likely to exert a beneficial influence on clinical symptom improvement in patients with EDs”. I agree, but not only would it not add anything, but it could potentially interfere with the therapeutic process (there is evidence that reducing anxious symptoms through drugs can reduce the habituation response to anxiety disorders). This is without counting the costs and side effects. Perhaps the authors will consider expanding this argument.

Response: Following the reviewer's recommendations, we considered it appropriate to review the bibliography and add a few sentences in the manuscript that would allow us to approach the subject with a broader perspective: “Despite this, other research has highlighted that the use of psychotropic drugs, such as benzodiazepines, may interfere negatively with some learning processes such as habituation. In addition, scientific evidence advises against the administration of anxiolytics and antidepressants in most cases because of their side effects and their limited long-term effect”. (p. 16, para. 2).

Comment #5: Also, in the discussion it is said that the treatment would be beneficial to increase emotional regulation. Although I believe that this statement may be true, the results of this study do not fully support this, since of the core symptoms, two of them have more to do with the physiological state (relaxation and rest) than with a cognitive process.

Response: following the reviewer's recommendations, we have drawn some conclusions more in line with the results of our study: “In our case, it is likely that the addition of TD-GCBT may help patients to decrease their physiological arousal and tolerate physical and emotional sensations (difficulty relaxing, restlessness and irritability), and to develop cognitive reappraisal strategies useful for changing the way an emotion is experienced and to generate more realistic interpretations (sad mood and worrying too much about different things)”. (p. 17, para. 1).

Comment #6: According to the results, it would seem that the most immediate benefit of transdiagnostic group therapy would be in anxiety symptoms and, subsequently, in depressive symptoms. It is argued that it could be the former who bring about change in the latter. In that case, would it be wise to focus exclusively on a treatment for anxiety disorders and hope that it would also have an effect on depressive symptoms? What would justify the use of a transdiagnostic approach then? Or is it the treatment itself that takes time to take effect? Would the intervention change depending on the core disorder or symptoms? It would be helpful for the authors to reflect on these issues and outline how treatments can be individualized to be useful to clinicians.

Response: We welcome the various reflection questions posed by the reviewer. First of all, we would like to point out that psychological treatment initially exerts a greater effect on certain anxiety symptoms compared to TAU. However, this does not indicate that transdiagnostic group therapy does not influence depressive symptoms. Therefore, and given that our study sample presents heterogeneous symptoms, we think that transdiagnostic therapy may be an appropriate intervention to ameliorate EDs. However, in the future it would be useful to compare specific interventions for anxiety or depression versus transdiagnostic approaches, and to observe differences in direct effects and in the evolution of network structure in people with EDs. On the other hand, we think that intervention techniques should be different depending on the core disorder or the most important symptoms. To approach this goal, it would be interesting, for example, to design N = 1 investigations and collect more measures during treatment, to see the direct associations that emerge after each treatment module and to design more personalized treatments.

Comment #7: In the limitations section, it would be convenient to add that these results are for mild and moderate symptoms, since they could change substantially when dealing with serious emotional disorders. Furthermore, the extent to which comorbidity between anxiety, depression, somatization and panic can influence the results is not studied. Finally, it would also be appropriate to comment that there could be differences between the group and individual treatment formats.

Response: Following the reviewer's suggestions, we added the following limitations to the manuscript: “Fourth, it is unlikely that the symptomatic variables of EDs selected in this study perfectly capture all variables relevant to understanding the effect of TAU or TD-GCBT + TAU on specific symptoms. Thus, future research should take into account the comorbidity between anxiety, depression, somatizations, and panic and include other variables of interest. Fifth, the findings of the present study can only be considered for mild and moderate symptoms”. (p. 20, para. 2). 

REVIEWER #3:

Comment #1: In the first paragraph of the introduction, the author mentions emotional disorders, highlighting depression, anxiety, and other related disorders. It would be beneficial for the author to specify the scope and definition of these categories to provide clarity to the readers. Defining the specific disorders encompassed within the term "other related disorders" will enhance the overall understanding of the study's context.

Response: Several authors have defined the concept of "emotional disorders" indicating which DSM diagnoses were included within this concept (anxiety and depressive disorders, somatoform disorders, panic disorder, phobias, bipolar disorder, obsessive-compulsive disorder). These disorders under the umbrella of emotional disorders appear to share underlying functional processes and meet a number of criteria: 1) the experience of frequent and intense negative emotions; 2) negative appraisal of the emotion and an aversive reaction to the emotional experience; and 3) the individual makes efforts to buffer, escape or avoid the emotional experience. However, as Bullis et al. (2019) point out, the term "emotional disorders" has been used primarily and more consistently in reference to anxiety and depressive disorders. In our manuscript, we have redefined it.: “Emotional disorders (EDs; mainly, anxiety and depressive disorders)”. (p. 3, para. 1). 

Bullis, J. R., Boettcher, H., Sauer‐Zavala, S., Farchione, T. J., & Barlow, D. H. (2019). What is an emotional disorder? A transdiagnostic mechanistic definition with implications for assessment, treatment, and prevention. Clinical Psychology (New York), 26(2), 1-19. https://doi.org/10.1111/cpsp.12278

Comment #2: Within the same paragraph, the manuscript presents statistics on the prevalence of depressive and anxiety disorders, citing data from a study conducted in 2017. Given that the manuscript also alludes to the potential exacerbation of these effects following the pandemic, it would be advisable to update these epidemiological figures. The COVID-19 pandemic has had a significant impact on mental health, and it is likely that the prevalence of these disorders has increased since 2017. Updating this information with more recent data would provide a more accurate and relevant context for the study.

Response: Following the reviewer's recommendations, we have updated the prevalence data for depressive and anxiety disorders.: “The COVID-19 pandemic increased the prevalence of anxiety and depression worldwide [2,3]. Specifically, depressive disorders increased by 27.6% and anxiety disorders by 25.6%”. (p. 3, para. 1).

Comment #3: While the Measures section delves into detailed descriptions of the instruments used in the study, it would be beneficial to specify this information in the Study Design and Procedure section as well. In the current text, there is a lack of detailed information regarding the questionnaires employed, with only a brief mention of the PHQ (Patient Health Questionnaire).

Response: Following the reviewer's suggestions, we have also added information from the questionnaires used in the study in section “Study design and procedure”: “Specifically, the Patient Health Questionnaire-9 (PHQ-9) [44] and the Generalized Anxiety Disorder-7 (GAD-7) were used to assess symptoms of depression and anxiety, respectively”. (p. 6-7). 

Comment #4: In the manuscript, it is mentioned that "Patients with PHQ scores indicative of major depression were re-evaluated by a clinical psychologist using a semi-structured interview". It would be interesting to clarify the specific cut-off point employed to determine the criteria for major depression. Furthermore, it would be beneficial to understand the basis of this interview and the specific criteria used for the inclusion or exclusion of participants in the study.

Response: Following the reviewer's recommendations, we have indicated the cut-off points used to determine the severity criteria for depression and anxiety (p. 7, para. 1), as well as the interview used by clinical psychologists to confirm the presence of a major depressive disorder or a major anxiety disorder (Structured Clinical Interview for DSM Axis-I Disorders; SCID-I) (p. 7, para. 1). The changes made to clarify these issues can be seen in the "Study design and procedure" section. 

Comment #5: It is my understanding that the presence of a major depressive disorder is considered an exclusion criterion in the study. However, the inclusion and exclusion criteria do not explicitly specify this information. Instead, they refer to the "presence of emotional symptomatology" without detailing the severity of these symptoms.

Response: We again refer the reviewer to the "Study design and procedure" section to observe the changes made with respect to the inclusion and exclusion criteria. 

Comment #6: Furthermore, in the Methods section, it is mentioned that "patients presented signs or symptoms of negative emotional problems, moderate depression, anxiety, or somatic symptoms." However, in the Discussion, reference is made to "mild-moderate EDs." This inconsistency in terminology could lead to confusion among readers regarding the study's eligibility criteria.

Response: We have revised the text and unified the terminology throughout the text indicating that our sample is represented by patients with mild-moderate clinical symptoms of an emotional disorder. 

Comment #7: Regarding the measures used in the study, it would be valuable to specify whether these instruments are self-administered by the participants.

Response: Following the reviewer's recommendations, we have specified in the manuscript that the questionnaires are self-administered by the patients: “The self-administered questionnaires included measures of clinical symptoms that were evaluated using the Patient Health Questionnaire (PHQ)”. (p. 6, para. 3).

Comment #8: I recommend verifying the data presented in Table 1. Specifically, in the "Marital Status" category, it appears that the number of individuals exceeds the sample size specified in the study.

Response: Thank you for your comment. We have reviewed the sociodemographic data and indeed we had made a mistake when transferring the data to the table. In the "unmarried" participants it indicated 260 and we have replaced it with the correct number which is 162. (p. 8, Table 1).

Comment #9: My understanding is that the experimental group combines TD-GCBT and TAU alone. It would be beneficial to provide details on how these treatments are combined. Is there a specific protocol with a defined number of TAU sessions? Can these treatments overlap in time?

Response: As suggested by the reviewer, we have provided more details in the manuscript on how the two treatments could be combined: “Patients receiving pharmacological treatment prior to the start of the study could also be randomly assigned to the experimental group. Once assigned, primary care physicians were not allowed to prescribe new medications or increase pharmacotherapy to these participants, but could decrease or eliminate medication if there was improvement”. (p. 10, para. 2). To clarify the questions posed by the reviewer: (1) There was no specific protocol with a defined number of TAU sessions, but rather each patient took the medication prescribed by his or her family physician, and (2) both treatments (pharmacological and psychological) could overlap in the case of participants who were previously taking medication, with the indication to family physicians that they could not prescribe new drugs or increase the dose (only decrease or withdraw the medication). 

Comment #10: Furthermore, it would be valuable to provide a concise description of the TD-GCBT protocol. The manuscript makes references to different modules in the discussion, but these modules are not explained in detail earlier in the text. Providing a brief overview of the treatment protocol will help readers understand the specific components and methodologies involved in TD-CBT, which is crucial for interpreting the study's findings and relevance.

Response: As suggested by the reviewer, we have provided a brief description of the components of the TD-GCBT protocol in the section "Description of treatments": “The TD-GCBT protocol modules are as follows: (1) introduction and psychoeducation: presentation and explanation of the protocol and information to participants about emotions, their adaptive function and when they become maladaptive and turn into EDs; (2) relaxation: reduction of psychophysiological activation through different self-regulation strategies (diaphragmatic breathing, progressive muscle relaxation and visualization); (3) Cognitive restructuring: information about rational and irrational thoughts and strategies to modify them; (4) behavioral therapy: behavioral activation, exposure techniques, social skills and problem solving and (5) relapse prevention: acceptance of relapses and restructuring of relapses”. (p. 10, para. 3).

Comment #11: During the various assessment time points, has there been any monitoring or control to ascertain whether patients engaged in other forms of psychological therapy beyond the established sessions? While I understand that not receiving any other psychological therapy during the established sessions is an exclusion criterion, it is not clear whether patients are permitted to pursue additional therapy during the follow-up period.

Response: Yes, after treatment and in the different follow-up measures, a question was included in the questionnaire battery to determine whether the participants were following another therapy in addition to the one established in the study. The question was: "Since the beginning of your participation in this study, have you received any other psychological or psychiatric therapy (public or private)?". Participants were asked to answer yes or no. If the answer was yes, the participant was automatically excluded from the study. (p. 7). 

Comment #12: In the Results section, there is a sentence that reads as follows: "Specifically, at follow-up, TD-GCBT+TAU was associated with a decreased..." My concern pertains to the term "follow-up." While it appears that this term may refer to post-treatment based on the data presented in the figure, it would be beneficial to explicitly clarify which specific measure of follow-up is being referenced. Ensuring this clarification will eliminate potential ambiguity and enhance the readers' understanding of your results.

Response: Following the reviewer's recommendations, we have rewritten some sentences of the results to reduce possible ambiguities and clarify the time point to which each differential treatment effect of the experimental group corresponds. (p. 14-15). 

Comment #13: I have noted that in the Discussion section, you mention the variation in the number of participants at different time points as a limitation of the study. I would like to obtain more information on this matter. Could you provide details on how many participants were considered at each assessment time point? Were any data imputation methods employed to address potential variations in sample size at different time points?

Response: In the statistical analysis section, we have provided information on the number of participants at each time point and explained how the data were imputed. (p. 11, para. 1). 

Comment #14: It would be interesting to specify in the discussion that the results cannot be generalized to other populations, as the one considered in the study is very specific.

Response: Following the reviewer's recommendations, we have specified in the discussion the following limitation: “Sixth, since this is a very specific sample, the results cannot be generalized to other populations”. (p. 21, para. 1).

Comment #15: The authors stated that "treatments appear to affect specific symptoms first and subsequently trigger a wave of changes in other symptoms indirectly, thus modifying the connections between network elements." However, how does the author explain the occurrence of these changes in variables like anhedonia or energy, but not in other variables such as sleep and appetite?

Response: The direct edges observed in the network indicate a greater effect of one of the treatments on those particular symptoms compared to the other treatment. However, the fact that no direct associations appear between the treatment variable and the symptoms does not indicate that there is no effect on those symptoms. These effects may exist, but, when a direct edge between the treatment variable and some symptom appears, it indicates that this treatment is superior to the other in modifying these symptoms. In our study, it is likely that the direct and differential effects in favour of TD-GCBT (red edges) are due to the specific techniques included in the protocol modules. For example, cognitive restructuring strategies or behavioral techniques (behavioral activation, exposure techniques, social skills and problem solving) could favour, in comparison with TAU, the improvement of depressed mood and anhedonia; then, these changes could facilitate the increase of activities and consequently of energy. Therefore, certain specific techniques could have differential effects on some specific symptoms, but not on others. (p. 17, para. 2). 

Comment #16: Similarly, there is an observed loss of association in variables like irritability at three and twelve months, but not at six months. A similar pattern is also noted with difficulty in relaxing. How can this variation be explained?

Response: We hypothesize that there could be different reasons for this temporal variability, as we reflected in the discussion: “On the other hand, it should be noted that there is temporal variability in the direct effects of TD-GCBT on some symptoms. For example, direct associations on irritability emerge at post-treatment and at 6 months, but not at 3 or 12 months follow-up. Similar results were observed in the difficulty to relax. Several reasons could explain such variability in the effects of TD-GCBT: first, the order established in the treatment modules; second, the variability between subjects in the training and in the regular practice followed by each participant after the end of the treatment and, finally, a statistical explanation based on the Regularization process (i.e., if there is a similar effect at two time points, the link between two nodes will appear exclusively in the one that presents statistical significance). However, this does not mean that there are no effects on these symptoms at the other time point, but that these have not been significant compared to the other time point”. (p. 18, para. 2). 

We thank the reviewers for the valuable comments they have provided and to give us the opportunity to substantially improve the manuscript for publication in the PLOS ONE.

---

## [Decision Letter · Decision Letter 1]

20 Mar 2024

Comparing psychological versus pharmacological treatment in emotional disorders: A network analysis

PONE-D-23-20048R1

Dear Dr. González,

We’re pleased to inform you that your manuscript has been judged scientifically suitable for publication and will be formally accepted for publication once it meets all outstanding technical requirements.

Kind regards,

Patricia Moreno-Peral

Academic Editor

PLOS ONE

Additional Editor Comments (optional):

Reviewers' comments:

Reviewer's Responses to Questions

**Comments to the Author**

1. If the authors have adequately addressed your comments raised in a previous round of review and you feel that this manuscript is now acceptable for publication, you may indicate that here to bypass the “Comments to the Author” section, enter your conflict of interest statement in the “Confidential to Editor” section, and submit your "Accept" recommendation.

Reviewer #1: All comments have been addressed

Reviewer #2: All comments have been addressed

Reviewer #3: All comments have been addressed

2. Is the manuscript technically sound, and do the data support the conclusions?

Reviewer #1: Yes

Reviewer #2: Yes

Reviewer #3: Yes

3. Has the statistical analysis been performed appropriately and rigorously? 

Reviewer #1: Yes

Reviewer #2: Yes

Reviewer #3: Yes

4. Have the authors made all data underlying the findings in their manuscript fully available?

Reviewer #1: Yes

Reviewer #2: (No Response)

Reviewer #3: Yes

5. Is the manuscript presented in an intelligible fashion and written in standard English?

Reviewer #1: (No Response)

Reviewer #2: Yes

Reviewer #3: Yes

6. Review Comments to the Author

Reviewer #1: I am satisfied with the authors' responses and the resulting changes to the paper.......................

Reviewer #2: (No Response)

Reviewer #3: I have carefully considered the feedback provided by the author and I am pleased to report that the author has addressed all the issues raised in the comments.

7. PLOS authors have the option to publish the peer review history of their article (what does this mean?). If published, this will include your full peer review and any attached files.

Reviewer #1: No

Reviewer #2: No

Reviewer #3: No

---

## [Editor Report · Acceptance letter]

25 Mar 2024

PONE-D-23-20048R1 

PLOS ONE

Dear Dr. Jurado-González, 

I'm pleased to inform you that your manuscript has been deemed suitable for publication in PLOS ONE. Congratulations! Your manuscript is now being handed over to our production team.

Kind regards, 

on behalf of

Dr. Patricia Moreno-Peral 

Academic Editor

PLOS ONE